Stability of an ophthalmic formulation of polyhexamethylene biguanide in gamma-sterilized and ethylene oxide sterilized low density polyethylene multidose eyedroppers

http://orcid.org/0000-0002-6937-1796 Bouattour Yassine 1
http://orcid.org/0000-0001-8086-7517 Chennell Philip 2 pchennell@chu-clermontferrand.fr
Wasiak Mathieu 1
Jouannet Mireille 1
Sautou Valérie 2
1 CHU Clermont-Ferrand, Pôle Pharmacie , Clermont-Ferrand , France
2 Université Clermont Auvergne, CHU Clermont-Ferrand, CNRS, SIGMA Clermont-Ferrand, ICCF , Clermont-Ferrand , France
Macknik Stephen
Electronic publication date: 2018 Apr 18
Publication date: 2018
Volume: 6
Electronic Location ID: e4549
Received 2018 Jan 29; Accepted 2018 Mar 8
Copyright: © 2018 Bouattour et al.
Copyright year: 2018
Copyright holder: Bouattour et al.
License: This is an open access article distributed under the terms of the Creative Commons Attribution License, which permits unrestricted use, distribution, reproduction and adaptation in any medium and for any purpose provided that it is properly attributed. For attribution, the original author(s), title, publication source (PeerJ) and either DOI or URL of the article must be cited.
License URL: https://creativecommons.org/licenses/by/4.0/

Keywords: Polyhexamethylene biguanide, Ophthalmic solution, Drug stability, Pharmaceutical technology, Acanthamoeba keratitis, Liquid chromatography, Low density polyethylene eye droppers

Funding: The authors received no funding for this work.

==============================
Background

Polyhexamethylene biguanide (PHMB) eye drops are a frequently used medication to treat Acanthamoeba keratitis. In the absence of marketed PHMB eye drops, pharmacy-compounding units are needed to prepare this much needed treatment, but the lack of validated PHMB stability data severely limits their conservation by imposing short expiration dates after preparation. In this study we aim to assess the physicochemical and microbiological stability of a 0.2 mg/mL PHMB eye drop formulation stored in two kinds of polyethylene bottles at two different temperatures.

Methods

A liquid chromatography coupled with diode array detector stability-indicating method was validated to quantify PHMB, using a cyanopropyl bonded phase (Agilent Zorbax Eclipse XDB-CN column 4.6 × 75 mm with particle size of 3.5 μm) and isocratic elution consisting of acetonitrile/deionized water (3/97 v/v) at a flow rate of 1.3 mL/min. PHMB eye drops stability was assessed for 90 days of storage at 5 and 25 °C in ethylene oxide sterilized low density polyethylene (EOS-LDPE) and gamma sterilized low density polyethylene (GS-LDPE) bottles. The following analyses were performed: visual inspection, PHMB quantification and breakdown products (BPs) screening, osmolality and pH measurements, and sterility assessment. PHMB quantification and BP screening was also performed on the drops emitted from the multidose eyedroppers to simulate in-use condition.

Results

The analytical method developed meets all the qualitative and quantitative criteria for validation with an acceptable accuracy and good linearity, and is stability indicating. During 90 days of storage, no significant decrease of PHMB concentration was found compared to initial concentration in all stored PHMB eye drops. However, BP were found at day 30 and at day 90 of monitoring in both kind of bottles, stored at 5 and 25 °C, respectively. Although no significant variation of osmolality was found and sterility was maintained during 90 days of monitoring, a significant decrease of pH in GS-LDPE PHMB eye drops was noticed reaching 4 and 4.6 at 25 °C and 5 °C respectively, compared to initial pH of 6.16.

Discussion

Although no significant decrease in PHMB concentration was found during 90 days of monitoring in all conditions, the appearance of BPs and their unknown toxicities let us believe that 0.2 mg/mL PHMB solution should be conserved for no longer than 60 days in EOS-LDPE bottles at 25 °C.

Introduction

Acanthamoeba keratitis (AK) is a rare but severe sight-threatening ocular infection with increasing prevalence worldwide. It is most common in contact lens (CL) wearers, with reported rates in the range of 1–33 cases per million CL wearers (Maycock & Jayaswal, 2016). Patients with AK may experience pain with photophobia, ring-like stromal infiltrate, epithelial defect and lid oedema. If AK is not treated adequately and aggressively, it can lead to loss of vision (Lorenzo-Morales, Khan & Walochnik, 2015). AK treatment must therefore be effective on both trophozoites and cyst stage of Acanthamoeba. Indeed, Acanthamoeba trophozoites are susceptible to antibiotics, antiseptics, antifungals and antiprotozoals. However, due to the existence of a cyst form in Acanthamoeba that is highly resistant to therapy, this may lead to prolonged or resistant infections, as most of the above treatments remain ineffective. The first line treatments of AK are biguanides such as chlorhexidine or polyhexamethylene biguanide (PHMB) (Lorenzo-Morales, Khan & Walochnik, 2015; Maycock & Jayaswal, 2016). Polyhexamethylene biguanide, also known as Polyhexanide PHMB is a polymer used for its antiseptic and disinfectant properties. Its activity is due to interactions with the phospholipid membrane of the infectious cell, caused by rapid attraction of PHMB toward the negatively charged bacterial cell surface, with strong and specific adsorption to phosphate-containing compounds causing impairment of the integrity of the outer membrane. PHMB is then attracted to the inner membrane and bonded to phospholipids, with an increase in inner membrane permeability accompanied by bacteriostasis; complete loss of membrane function follows, with precipitation of intracellular constituents and a bactericidal effect (Broxton, Woodcock & Gilbert, 1983; Ikeda, Tazuke & Watanabe, 1983; Ikeda et al., 1984; Broxton et al., 1984). Due to this mechanism, PHMB is effective against numerous organisms like viruses such as HPV (Gentile, Gerli & Di Renzo, 2012) or HIV (Krebs et al., 2005; Passic et al., 2010), Gram-negative and Gram-positive bacteria (Yanai et al., 2011), and fungi. In 1992, Larkin et al. first reported the success of treating five of six patients with a 0.02% eye drops of PHMB, qualifying the treatment as promising in this infection (Larkin, Kilvington & Dart, 1992). A retrospective evaluation of the clinical outcomes of AK patients treated with unlicensed drugs, realized by the Community Research and Development Information Service (CORDIS), indicated that 96% of patients were using CLes, and that PHMB was the most frequently used medication. To treat AK, PHMB eye drops are given hourly (day and night) for the first 48 h before being reduced to hourly (daytime only) for several days or weeks. It is important that the treatment regimen is reduced and tailored to each clinical case to minimize any epithelial toxicity. The aim is to reduce therapy to four times a day, but patients may need to be on therapy for up to six months (Radford, Lehmann & Dart, 1998).

The efficacy of PHMB established it as a promising candidate for obtaining an official AK treatment license by the European Medicines Agency (Community Research and Development Information Service (CORDIS), 2015). A phase I study of PHMB eye drops has already been completed and a phase III is still ongoing to demonstrate experimental scientific evidence on the quality, safety and efficacy of PHMB and to prepare the basis for a Marketing Authorization application. (PHMB Ophthalmic Solution in Subjects Affected by AK; SIFI SpA (2017)).

Until then, pharmacy-compounding units have to prepare PHMB eye drops, but the lack of long-term validated PHMB stability data severely limits their conservation period by imposing short expiration dates after preparation.

The objective of this study was to assess the physicochemical stability and to control the sterility of a PHMB 0.2 mg/mL (0.02%) ophthalmic solution in two different low-density polyethylene multidose eyedropper, at two conservation conditions (5 °C and 25 °C) for three months in unopened eyedroppers and during 10 h of simulated patient use.

Materials and Methods

Preparation and storage of PHMB solution formulations

A total of 500 mL of 0.2 mg/mL solutions of PHMB were prepared by diluting 500 μL of 200 mg/mL PHMB solution (batch 15H28-B11, expiring September 2017, Fagron Services, Netherlands) in 49.5 mL of deionized water (Versylene®; Fresenius Kabi France, Louviers, France) to obtain a 2 mg/mL solution, which was then diluted with 450 mL of sterile 0.9% Sodium chloride solution (Versylene®; Fresenius Kabi France, Louviers, France).

The solutions were sterilely distributed (2 mL per unit) using a sterile syringe tipped with a 0.22 μm pore size filter (reference SLGP033RS Millipore SAS batch number R5MA47234; Millipore, Molsheim Cedex, France) under the laminar air flow of an ISO 4.8 microbiological safety cabinet into two different multidose low density polyethylene (LDPE) eyedroppers: Gamma sterilized white opaque LDPE (GS-LDPE) eyedropper squeezable bottle (reference 10002134 with Novelia® caps reference 20050772; Nemera, La Verpillière Cedex, France).

Ethylene oxide sterilized white opaque LDPE (EOS-LDPE) eyedropper squeezable bottle (reference VFLA10B10; Laboratoire CAT®, Montereau en Gatinais, France).

Study design

The stability of 0.2 mg/mL PHMB solution was studied in the two kinds of unopened multidose eyedroppers for 90 days at 25 °C or 5 °C, and in simulated use conditions for 10 h.

Stability of 0.2 mg/mL PHMB solution in unopened multidose eyedroppers

The eyedroppers containing PHMB were stored upwards at controlled refrigerated temperature (Whirlpool refrigerator) at 5 °C ± 3 °C or in a climate chamber (BINDER GmbH, Tuttlingen, Germany) at 25 °C ± 2 °C and 60% ± 5% residual humidity, until analysis.

Immediately after preparation, and at day 4, 8, 15, 30 and 90, three units per kind of eyedropper and storage temperature were subjected to the following analyses: visual inspection, PHMB quantification, breakdown products (BPs) research, osmolality and pH measurements. Sterility was also assessed using three units for each kind of eyedropper and storage temperature immediately after preparation and after 30 and 90 days of storage.

As a complementary study, pH monitoring of sodium chloride 0.9% solutions was also realized in coloured gamma sterilized LDPE bottle (reference BW-F180; Nemera®, La Verpillière Cedex, France), coloured non-sterilized LDPE bottles (reference BW-F176; Nemera®, La Verpillière Cedex, France), translucent non-sterilized LDPE bottles (reference BW-F178; Nemera®, La Verpillière Cedex, France) and translucent non-sterilized polypropylene (PP) bottles (reference C52116583658; Nemera®, La Verpillière Cedex, France), for 30 days, at 25 °C and 60% residual humidity.

Evaluation of PHMB concentrations in eye drops during simulated use

At day 0, 10 eyedroppers for each tested condition were subjected to simulated patient use: every hour for 10 h, a drop was emitted, and PHMB quantification was realized in 10 pooled drops of the 10 bottles per conditioning and per storage temperature condition.

Analyses performed on the PHMB solutions

Visual inspection

The multidose eyedroppers were emptied into polycarbonate test tubes and the PHMB solutions were visually inspected under day light. Aspect and colour of the solutions were noted, and a screening for visible particles, haziness, or gas development was performed.

PHMB quantification and BPs research

Chemicals and instrumentation

For each unit, PHMB was quantified and BPs detected using a stability-indicating method adapted from Küsters et al. (2013), by liquid chromatography (LC) using AS-2055 Plus sampler with LG-1580-02 pump, Jasco MD-2018 plus with diode array detector in ultraviolet-visible range (DAD UV-Vis) and ChromNAV® software integrator (Jasco, Evry, France). The LC separation column used was an Agilent Zorbax Eclipse XDB-CN column (4.6 × 75 mm) with particle size of 3.5 μm and guard column (4.6 × 12.5 mm). The mobile phase was an acetonitrile/water mixture (3/97% v/v) in which the acetonitrile was HPLC quality (Chromasolv® for HPLC; Honeywell®, Roissy CDG, France) and the water was deionized and sterile (Versylene®; Fresenius Kabi France, Louviers, France). The flow rate through the column for the analysis was set at 1.3 mL/min, with the column thermo-regulated to a temperature of 25 °C. The injection volume was 100 μL. The quantification wavelength was set up at 220 nm. BP detection was realized by screening with DAD detector from 200 to 600 nm.

Validation method

Linearity was initially verified by preparing one calibration curve daily for three days using five concentrations of PHMB 0.12, 0.16, 0.2, 0.24 and 0.28 mg/mL. Each calibration curve should have determination coefficient R2 equal or higher than 0.999. Homogeneity of the curves was verified using Cochran test. ANOVA test was applied to determine applicability of linear model.

Each day for three days, six solutions of PHMB 0.2 mg/mL were prepared, analysed and quantified using a calibration curve prepared the same day. To verify the method precision, repeatability was estimated by calculating relative standard deviation (RSD) of intraday analysis and intermediate precision was evaluated using RSD of inter-day analysis. Both RSDs should be less than 5%. Specificity was assessed by comparing UV spectra using DAD detector. Method accuracy was verified by evaluating recovery of five theoretical concentrations to experimental values found using mean curve equation, and results should be found within the range of 95–105%. Limit of detection (LOD) and limit of quantification (LOQ) were estimated based on the standard deviation of the response and the slope, following ICH guidelines. For three days, LOD was verified based on visual inspection of the chromatograms of six samples and LOQ was verified by calculating repeatability and trueness of measurement on 10 samples.

In order to exclude potential interference of degradation products with PHMB quantification, PHMB 0.2 mg/mL solutions was subjected to many forced degradation conditions. Degradation conditions were the following: 0.5 N hydrochloric acid for 30 min at 80 °C, 0.5 N sodium hydroxide for 120 min at 80 °C, 3% hydrogen peroxide at 80 °C for 24 h and thermal degradation for 24 h at 80 °C. Susceptibility to light was not performed due to use of opaque bottles because of known photostability of PHMB (Roth et al., 2011).

Osmolality and pH measurements

For each unit, pH was measured using a SevenMulti TH S40–Mettler Toledo pH-meter equipped with InLab® Micro Pro pH electrode. Measures were preceded and followed by instrument validation using standard buffer solution of pH 4 (HANNAH® Instrument, Tannerries, France).

Osmolality was measured for each solution using an osmometer Model 2020 Osmometer® (Radiometer, SAS; Advanced instruments Inc., Neuilly Plaisance, France).

Sterility assay

Sterility test method was validated (i.e. growth promotion test, validation of the sterility assay and elimination of the inhibitory effect) by Thibert et al. (2014), using a method adapted from the European Pharmacopoeia sterility assay (2.6.1). Multidose eyedroppers were opened under the laminar air flow of an ISO 4.8 microbiological safety cabinet, and the contents filtered under vacuum using a Nalgene analytical test filter funnel onto a 47 mm diameter cellulose nitrate membrane with a pore size of 0.45 μm (ref 147-0045, Thermo Electron SAS; ThermoFisher Scientific, Courtaboeuf Cedex, France). The membranes were then rinsed with 90 mL of Letheen broth (VWR International, Pessac, France), to remove any antibacterial effect of the solution and divided into two equal parts. Each individual part was transferred to each of a fluid thioglycolate and soya tripcase medium, and incubated at 30–35 °C or 20–25 °C respectively, for 14 days. The culture medium was then examined for colonies.

Data analysis—acceptability criteria

The stability of diluted PHMB solutions was assessed using the following parameters: visual aspect of the solution, PHMB concentration, presence or absence of BPs, pH and osmolality.

The study was conducted following methodological guidelines issued by the International Conference on Harmonisation for stability studies (ICH guidelines for stability—International Conference of Harmonization (ICH), 1996–2006; International Council for Harmonisation of Technical Requirements for Pharmaceuticals for Human Use (ICH), 1995), and recommendations issued by the French Society of Clinical Pharmacy (SFPC) & Evaluation and Research Group on Protection in Controlled Atmosphere (GERPAC) (2013). A variation of concentration outside the 90–110% range of initial concentration (including the limits of a 95% confidence interval of the measures) was considered as being a sign of instability. Presence of BPs and the variation of the physicochemical parameters were also considered a sign of PHMB instability. The observed solutions must be limpid, of unchanged colour, and clear of visible signs of haziness or precipitation. Since there are no standards that define acceptable pH or osmolality variation, pH measures were considered to be acceptable if they did not vary by more than one pH unit from initial value (French Society of Clinical Pharmacy (SFPC) & Evaluation and Research Group on Protection in Controlled Atmosphere (GERPAC), 2013), and osmolality results were interpreted considering clinical tolerance of the preparation.

Results

PHMB quantification and BPs research

PHMB retention time was of 9.1 ± 0.8 min. The chromatographic method used was found linear for concentrations ranging from 0.12 to 0.28 mg/mL with a determination coefficient R2 equal or higher to 0.999. Average regression equation was y = 3985934.2x − 12276.9 where x is the PHMB concentration and y the surface area of the corresponding peak. Interception was not significantly different from zero and average determination coefficient R2 of three calibration curves was 0.988. Recovery of 0.2 mg/mL was 98.85 ± 0.005, repeatability’s RSD was 2.2% and intermediate precision’s RSD was 3%. LOD was equal to 0.021 mg/mL and LOQ was 0.065 mg/mL, and found within the range of linearity of the calibration curve with recovery coefficients of less than 5%.

Three impurities were found to be already present in the PHMB solution, two of them being visible only at 200 nm and with diode array detection, showing retention times of 3.0, 3.5 and 7.7 min on reference chromatograms and relative retention time of 0.32, 0.38 and 0.84 compared to PHMB peak (Fig. 1). At 200 nm, the area of impurities one and three represented respectively 0.51%, and 1% of PHMB peak area whereas only impurity two was visible at 220 nm, representing 0.68% of PHMB peak surface.

Figure 1 Reference chromatograms of 0.2 mg/mL PHMB solution.

(A) At 200 nm; (B) at 220 nm and (C) diode array detection screening.

After forced degradation (Fig. 2), several BPs were detected with a resolution higher than 1.5 of PHMB peak to all its BPs, except for thermal degradation which showed no change in chromatograms after 24 h of heating at 80 °C. Figure 3 summarises relative retention times and intensities of all detected impurities and BPs function of the degradation conditions.

Figure 2 Screening with diode array detector of 0.2 mg/mL solution of PHMB after acid (A), alkaline (B), thermal (C) and oxidative (D) degradation.

Figure 3 Relative retention times and relative intensity of all detected impurities and breakdown products function of the degradation conditions at 220 nm.

*At 200 nm.

Stability of PHMB in unopened multidose eyedroppers

Physical stability

All samples stayed limpid and uncoloured during the study, for both tested concentrations and conservation temperatures, and there was no appearance of any visible particulate matter, haziness or gas development.

Chemical stability

Throughout the dosage times, mean concentrations of PHMB in all studied conditions did not vary by more than 5.2% of mean initial concentration (see Table 1).

Table 1 Evolution of PHMB concentration for each storage condition and kind of bottle in unopened eyedroppers (mean ± confidence interval, n = 3).

Kind of bottle and storage temperature	Initial concentration at Day 0 = 100% (mg/mL) (mean ± IC 95%)	% Of initial T0 measurement (mean ± CI 95%)	
Day 4	Day 8	Day 15	Day 30	Day 60	Day 90	
EOS-LDPE	
5 °C	0.204 ± 0.002	99.8 ± 3.19	98.8 ± 6.55	98.44 ± 5.2	103.91 ± 7.27	97.61 ± 13.37	96.42 ± 3.90	
25 °C	102.03 ± 4.48	98.94 ± 1.54	99.85 ± 4.90	101.21 ± 7.08	99.85 ± 4.63	97.58 ± 7.72	
GS-LDPE	
5 °C	0.201 ± 0.002	101.45 ± 10.99	104.5 ± 0.75	103.37 ± 8.9	102.92 ± 3.46	97.55 ± 15.43	97.87 ± 3.68	
25 °C	100.83 ± 7.63	99.55 ± 4.32	99.75 ± 0.38	103.62 ± 9.41	97.51 ± 11.06	95.89 ± 3.33	

Chromatographs showed no sign of BPs until day 60 included for both types of LDPE eyedroppers at 25 °C. After 90 days of storage, many BPs appeared including a BP which had not been detected during forced degradation, with a retention time of 2.5 min. An increase in intensity of the acid BP (retention time 1.5 min), was also noted for GS-LDPE bottles but not for EOS-LDPE bottles (Fig. 4). When stored at 5 °C, another previously undetected BP with a maximum of absorbance at 280 nm, appeared after 30 days, for both LDPE eyedroppers (Fig. 5).

Figure 4 Chromatogram of PHMB solution in GS bottles at 25 °C at day 0 (blue) and day 90 (red) in wavelength of 200 nm (A), 220 nm (B) and diode array detector screening (C).

Figure 5 Chromatograms of 0.2 mg/mL PHMB solution in GS-LDPE bottle after 30 and 90 days storage at 5 °C, viewed with 280 nm (PHMB not detectable at this wavelength) (A) and diode array detector screening (B).

BP, Breakdown products.

Throughout the study, osmolality did not vary by more than 1.94% (5 mosm/kg) of initial osmolality (258 mosm/kg) for both types of LDPE eyedroppers at both storage temperatures.

For the PHMB kept in EOS-LDPE, pH did not vary throughout the study by more than 0.52 and 0.46 pH units from initial pH (6.27) when stored at 5 °C and 25 °C respectively. However, the pH decreased considerably in the solutions conditioned in GS-LDPE, losing respectively 1.56 and 2.16 pH units from initial pH (6.16) after 90 days when stored at respectively 5 °C and 25 °C (see Fig. 6).

Figure 6 pH evolution in PHMB 0.2 mg/mL solution during 90 days (mean ± SD, n = 3).

Straight red lines indicate initial pH ±1 unit.

Figure 7 shows no significant decrease of non-coloured non-sterilized PP bottle and coloured and non-coloured non-sterilized LDPE bottle. However, we found a major decrease of pH (over 2.1 unit) after 30 days in coloured gamma sterilized LDPE bottle.

Figure 7 pH monitoring of 0.9% NaCl in polypropylene, coloured gamma sterilized, coloured non-sterilized and non-coloured non-sterilized low density polyethylene bottles during 30 days.

Sterility assay

None of the three analysed solutions conserved in unopened bottles at day 0, day 30 or day 90 showed any signs of microbial growth.

Stability of PHMB in simulated use conditions

During 10 h of drop sampling, no variation exceeding of ±10% of initial concentration was found for any of the studied conditions, as presented in Table 2.

Table 2 Evolution of PHMB concentration (% of initial concentration) in drops for each storage condition and kind of bottle in in-use condition during 10 h, pooled from 10 bottles (n = 1).

	GS-LDPE	EOS-LDPE	
Initial concentration (mg/mL)	0.201	0.204	
	Pourcentage of initial concentration	
Hours	25 °C (%)	5 °C (%)	25 °C (%)	5 °C (%)	
0	99.3	98.4	103.9	103.9	
1	104.2	103.9	*	105.2	
2	99.7	103.6	101.3	105.1	
3	96.3	100.0	101.6	98.0	
5	105.9	105.7	105.7	100.1	
6	98.1	104.2	99.5	102.2	
7	99.7	104.4	103.1	104.8	
8	101.8	102.6	103.4	105.0	
9	102.7	102.9	101.3	101.9	
10	103.3	102.7	103.6	104.4	
Note:

* Not determined.

Discussion

Our study presents new data on the physicochemical stability of an aqueous formulation of PHMB conditioned in two differently sterilized LDPE eyedroppers (gamma radiations and ethylene oxide). Our study also highlighted the influence of the mode of sterilization of the eyedroppers on the properties of the solutions they are designed to contain, as well as identifying a surprising chemical degradation of PHMB when conserved in refrigerated conditions.

Initial chromatograms of PHMB 0.2 mg/mL showed the presence of three impurities, which is in accordance with the analysis certificate of the raw material, which reports the presence of four impurities, one of which (hexamethylenediamine) is not visible in UV (Pubchem, 2005). PHMB retention time presented variabilities with an RSD of 8.8% due to the high sensibility of the method to minute acetonitrile concentration variation. The use of DAD detector added more specificity to the method and allowed the identification of PHMB during analyses despite the slight variations of retention time.

Forced degradation assays showed that PHMB remained stable against most tested stress conditions except acidity. In fact, we found that PHMB concentration decrease was of about 28% after 15 min in 0.5 N of hydrochloric acid at 80 °C, 20.2% after 120 min in 0.5 N of NaOH solution at 80 °C and 12.3% after 24 h in 3% solution of hydrogen peroxide at 80 °C. On the other hand, PHMB decrease was only of 2% after 24 h of heating at 80 °C showing a resistance towards heating. All BPs appeared during forced degradation were sufficiently separated from PHMB peak with a resolution higher than 1.5. Since it was impossible to quantify BPs and impurities, we estimated the relative absorbance of those products compared to PHMB peak. All of the products have a relative absorbance between 0.4% and 5%, except for alkaline condition which showed two BPs of 68% and 118%, suggesting that those BPs have a higher molar extinction coefficient than PHMB.

During three months of monitoring, mean concentration of PHMB remained within a 90–110% range of initial concentration in EOS-LDPE and GS-LDPE bottles, at 5 and 25 °C. In fact, all the analysed solutions showed a decrease of less than 6% at the end of the study. However, variability of PHMB concentrations was noticed during monitoring, causing enlargement of confidence interval in EOS-LDPE eye drops stored at 5 °C, and in GS-LDPE stored at 5 and 25 °C, possibly suggesting the beginning of an instability of the drug at these conditions. Only PHMB solution in EOS-LDPE bottles stored at 25 °C stayed within the 90–110% of initial concentration range, when considering the 95% confidence interval. BPs began to appear at day 30 for solutions stored at 5 °C in both kind of bottles. Such results have not been reported in any research, but concurring information was found in the certificate of analyses of PHMB 20%, suggesting that solution storage should be at a temperature higher than 8 °C. Eye drops stored at 25 °C showed the appearance of several BPs at day 90 of monitoring for both kind of bottles as well, demonstrating that PHMB is chemically more stable at 25 °C than at 5 °C.

Although visual aspect was found unchanged and osmolality did not vary during the 90 days of monitoring in both conditioning at both temperatures, pH on the other hand presented a remarkable decrease during conservation in GS-LDPE bottles at 5 and 25 °C. In fact, we found that pH decreased from 6.16 to reach 4 and 4.6 at 25 °C and 5 °C respectively in GS-LDPE bottles, but no significant variation noticed for EOS-LDPE. Gamma rays are energetic rays that oxidise LDPE, and could result in liberation of protons that could acidify contained solution. To further verify the cause of pH decrease (LDPE, LDPE dye, LDPE method of sterilisation or the PHMB solution itself), pH comparison of 0.9% sodium chloride solution conditioned in sterilized coloured, non-sterilized coloured and non-coloured LDPE bottles, and uncoloured non-sterilized PP bottles was realised as well. pH comparison demonstrated that gamma sterilised eyedroppers were at risk of a pH decrease in sodium chloride solution and thus could explain pH decrease in PHMB solution. This is probably due to liberation of acids from LDPE when sterilized with gamma irradiation. In fact, Kawamura (2004) reported that acids such acetic acid, propionic acid, butanoic acid, pentanoic acid and acetone were detected in polyethylene and PP products after irradiation doses of 10, 30 and 50 kGy.

Six BPs was found at day 90 in both kind of LDPE eye drop bottles, among which one BP was not found in any condition during forced degradation assay, and another found in acid degradation conditions. The acid BP peak was very close to the solvent front peak which makes it difficult to quantify, but by comparing the intensity of this peak between solution in GS-LDPE and EOS-LDPE, we found that this peak was higher in GS-LDPE solution, which suggest that its formation was accelerated due to the acidification of the solution in GS-LDPE bottle. Since best stability of PHMB is between five and seven (Küsters et al., 2013), the acidification of the solution in GS-LDPE could therefore accelerate its degradation.

PHMB eye drops was described to be stable for 45 days when frozen and for 14 days when refrigerated if prepared with 0.01% benzalkonium chloride water (McElhiney, 2013). The difference between stabilities of those eye drops to ours could be explained by the use of different excipient, since we used 0.9% sodium chloride instead, considered as more isotonic, or alternatively the study was not performed for a longer time. Despite there being no other published data on ophthalmic formulations of PHMB, its stability was assessed in other pharmaceutical forms. For instance, Lucas et al. found that PHMB at 1 ppm in multipurpose CLes solution being stable after 30 days stored at −20 °C, 5 °C and 25 °C (Lucas, Gordon & Stratmeyer, 2009). However, since their quantification method were not stability indicating, it is hard to confirm the absence of potentially toxic BP in PHMB solutions. On the other hand, Küsters et al. (2013) also found PHMB at 0.02% stable for 12 months at 22 °C in wound gel and irrigation solution, but as previously mentioned, their method might not have been optimal for the detection of specific BPs and the galenic form of their product does not make it suitable for eye delivery.

Our data has proven that PHMB concentrations remained physicochemically stable throughout the study at 25 °C for up to two months without any detectable BPs appearing or other parameter modification when stored in EOS-LDPE bottles. This information is therefore in favour of identical quality of the product after 60 days of storage. A study of PHMB biological activity, using bioassays such as MIC testing or time-kills against standard bacterial isolates or the use of an Acanthamoeba complete-kill assay (Kowalski et al., 2013) could however confirm this information: as intact PHMB already possessed antimicrobial activity, unchanged PHMB should retain this activity. Such an additional study could however be used to investigate the activity of physicochemically unstable PHMB solutions. For example, when BPs are observed is the formulation still potent? Do these products or other compounds released from the container modify PHMB activity or affect ophthalmic tolerance? This investigation could therefore be coupled to a cytotoxicity study evaluating ‘unstable’ formulation, in order to assess the pertinence of the stringent quality specifications adopted for stability studies.

The sterility assay performed in our work followed the European Pharmacopeia sterility monography (Council of Europe & European Pharmacopoeia, 2014) and did not reveal any microbial contamination after three months of storage. Absence of microbiological growth during use is fundamental to improve patient safety. Optimal microbiological sterility can be achieved using single dose eyedroppers, however such a technology is not always applicable for all drugs or formulations and not available in most of hospital compounding department.

During simulated use of PHMB eye droppers, we evaluated PHMB concentrations in emitted eye drops for 10 h, in order to research any sorption phenomenon that could lead to a decrease of PHMB concentration in the delivered drops. Since PHMB eye drops must be taken each hour for the first 24 h, any loss of PHMB concentration during use may probably delay healing. Any sorption phenomenon that could have happened would have appeared early, so the absence of any concentration decrease leads us to believe that no clinically relevant sorption happened. In our experiments, we pooled 10 drops for analytical reasons. We found no significant variation of PHMB concentration in both LDPE bottles, especially for GS-LDPE which contain silicone in their caps, thus establishing that there is no clinically significant sorption between PHMB and the used eyedroppers.

Conclusion

Although no significant decrease in PHMB concentration was found during 90 days of monitoring, the appearance of BPs and their unknown toxicities let us believe that 0.2 mg/mL PHMB solution should be conserved for no longer than 60 days in EOS-LDPE bottles at 25 °C.

Supplemental Information

Supplemental Information 1 PHMB breakdown and impureties monotoring data.

Click here for additional data file.

Supplemental Information 2 PHMB concentration monotoring data.

Click here for additional data file.

Supplemental Information 3 Osmolality monitoring data.

Click here for additional data file.

Supplemental Information 4 pH monitoring data.

Click here for additional data file.

Supplemental Information 5 Complementory pH monitoring data.

Click here for additional data file.

Supplemental Information 6 PHMB alkalin degradation chromatograms.

Click here for additional data file.

Supplemental Information 7 PHMB oxydative degradation chromatograms.

Click here for additional data file.

Supplemental Information 8 PHMB acid degradation chromatograms.

Click here for additional data file.

Supplemental Information 9 PHMB thermal degradation chromatograms.

Click here for additional data file.

Additional Information and Declarations

Competing Interests

Author Contributions

Data Availability

The authors declare that they have no competing interests.

Yassine Bouattour conceived and designed the experiments, performed the experiments, analysed the data, contributed reagents/materials/analysis tools, prepared figures and/or tables, authored or reviewed drafts of the paper, approved the final draft.

Philip Chennell conceived and designed the experiments, analysed the data, contributed reagents/materials/analysis tools, prepared figures and/or tables, authored or reviewed drafts of the paper, approved the final draft.

Mathieu Wasiak conceived and designed the experiments, authored or reviewed drafts of the paper, approved the final draft, technical help.

Mireille Jouannet conceived and designed the experiments, authored or reviewed drafts of the paper, approved the final draft.

Valérie Sautou conceived and designed the experiments, contributed reagents/materials/analysis tools, authored or reviewed drafts of the paper, approved the final draft.

The following information was supplied regarding data availability:

The raw data has been supplied as Supplemental Files.

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
