# Peer review of "Stability of an ophthalmic formulation of polyhexamethylene biguanide in gamma-sterilized and ethylene oxide sterilized low density polyethylene multidose eyedroppers"

_PeerJ, doi:10.7717/peerj.4549_

## Round 0.1 · original submission · Minor Revisions

The reviewers were happy with the quality of the manuscript with the only significant comment being to add bioassays to test the biological activity (not just stability as measured). Reviewer 1: Bioassays, such as MIC testing or time-kills against standard bacterial isolates or the use of an Acanthamoeba complete-kill assay (Kowalski RP, Abdel Aziz S, Romanowski EG, Shanks RMQ, Raju LV. Development of a Practical Complete-Kill assay to Evaluate Anti-Acanthamoeba Drugs. JAMA Ophthalmol 2013;131(11):1459.) to determine bioactivity would have provided important supplemental data for this study. I agree this would be wise. Please address this concern in your resubmission.

Reviewer 1 ·

Basic reporting

The manuscript is written clearly and in proper English. The Introduction provides a large amount of background information which is cited appropriately. Figures and tables are appropriate and the raw data is supplied.

Experimental design

The research question is well defined and relevant. However I believe another measure of stability should have been performed. As a microbiologist, I believe that a bioassay demonstrating antimicrobial activity against a number of bacteria or Acanthamoeba isolates should have been performed to demonstrate antimicrobial similar activity at the various time points.

Validity of the findings

I believe the data is valid, but lacking antimicrobial biological activity data.

Additional comments

I commend the authors for a well-written manuscript describing the stability of PHMB eye drops over three months and contained in bottles sterilized using different methods and at different storage temperatures. A number of assays were used to analyze the PHMB concentrations over time, osmolality, pH, and breakdown products over time. The concentration of PHMB was determined using LC. While I believe that determining the PHMB concentrations and breakdown products over time is very important to demonstrated chemical stability, the authors do not demonstrate that the biological activity of PHMB was not affected. Bioassays, such as MIC testing or time-kills against standard bacterial isolates or the use of an Acanthamoeba complete-kill assay (Kowalski RP, Abdel Aziz S, Romanowski EG, Shanks RMQ, Raju LV. Development of a Practical Complete-Kill assay to Evaluate Anti-Acanthamoeba Drugs. JAMA Ophthalmol 2013;131(11):1459.) to determine bioactivity would have provided important supplemental data for this study. I believe that stability testing of all antimicrobials should be tested against susceptible and resistant organisms to determine whether there is any degradation of biological activity. If the authors do not have this data to include in this manuscript , then this point should be discussed as a limitation of the study in the manuscript and the reason why it was not done also included.

Minor Points
Line 73: It is stated that Acanthamoeba trophozoites are “sensitive” to antibiotics. Microorganisms are not “sensitive” to antibiotics, they are “susceptible”.

Line 88: The Gram in Gram-positive and Gram-negative should be capitalized as Gram is a proper name.

Lines 109 - 112: It is stated on lines 109-112 “The objective of this study was to assess the physicochemical and microbiological stability of a PHMB 0.2 mg/mL (0.02%) ophthalmic solution in two different low-density polyethylene multidose eyedropper, at two conservation conditions (5 °C and 25°C) for three months in unopened eyedroppers and during 10 hours of simulated patient use. It is stated that the “microbiological stability” of PHMB was assessed. Reading this, I am expecting to see microbiological data such as MICs, and not sterility data of the solutions. This statement should be reworded as the readers do not misinterpret the intent of the statement, so as to expect antimicrobial data rather than sterility of the solutions over time.

Reviewer 2 ·

Basic reporting

See uploaded PDF file

Experimental design

See uploaded PDF file

Validity of the findings

See uploaded PDF file

Additional comments

See uploaded PDF file

Annotated reviews are not available for download in order to protect the identity of reviewers who chose to remain anonymous.

---

## Round 0.2 · accepted · Accept

Thanks for your updates and revisions. I recommend publication.